# Expanding Computation Spaces of LLMs at Inference Time

## Abstract

Chain-of-thought (CoT) rationale enables language models to use additional task-related text for problem-solving, benefiting not only from detailed reasoning steps but also from the expanded computational space of longer inputs. Prior work has trained filler or special tokens to serve as additional computation spaces. In this study, we investigate whether language models can leverage artificially inserted sequences of filler tokens solely at inference. We first identify effective token types, numbers, and insertion locations, then examine at what stage of training models begin to exploit the expanded computation space, and finally analyze dynamics within these spaces via attention maps. Experiments on models ranging from 1.7B to 32B across open-domain QA and math tasks show that appropriate token types and counts vary, but placing filler tokens directly before the final 'Answer:' token is most effective. Smaller models benefit most, up to 12.372 percentage points in SmolLM2-1.7B-Instruct, indicating that these spaces act as additional computational capacity rather than redundant input. Attention maps reveal that expanded spaces often continue the original attention mechanism and sometimes focus on questions or answer options, suggesting meaningful computation for problem-solving.

## 1 Introduction

Chain-of-thought (CoT) prompting has been shown to substantially improve reasoning performance across tasks by guiding models to decompose and solve problems step by step, thereby making reasoning trajectories explicit (Hua & Zhang, 2022; Wei et al., 2022; Wang et al., 2022; Zelikman et al., 2024b). While its effectiveness partly stems from the detailed solution steps provided in the input, it has been hypothesized that longer inputs also help by providing a larger computational space. To investigate whether models indeed exploit this additional space, prior studies have introduced sequences of seemingly meaningless tokens, rather than CoT text, into the input (Herel & Mikolov, 2024; Goyal et al., 2023; Lanham et al., 2023). For example, they inserted repeated filler characters (e.g., '......') or special tokens (e.g., `<pause>`) at various positions in the input, thereby expanding the input length, and trained the model to leverage this extended space for problem-solving.

Extending this line of inquiry, we aim to study whether current language models are able to exploit the computation spaces from inserted tokens even without training. Our study explores whether tokens naturally present in the training corpus, and thus familiar to the models (e.g., *period*, *dash*, etc), can serve to enhance their problem-solving ability at inference time. Through pilot studies, we have found that even without explicitly training tokens in a particular form, providing additional tokens in the input as an *expanded computation space* can enhance model performance.

In this regard, we study three research questions in this work:

**RQ1.** What types and numbers of tokens are effective, and which parts of the input location benefit most from their insertion?

**RQ2.** When during training do models start to effectively exploit the expanded computation spaces to support answer inference?

**RQ3.** How do the extended token spaces interact with the original inputs and affect the answer prediction?

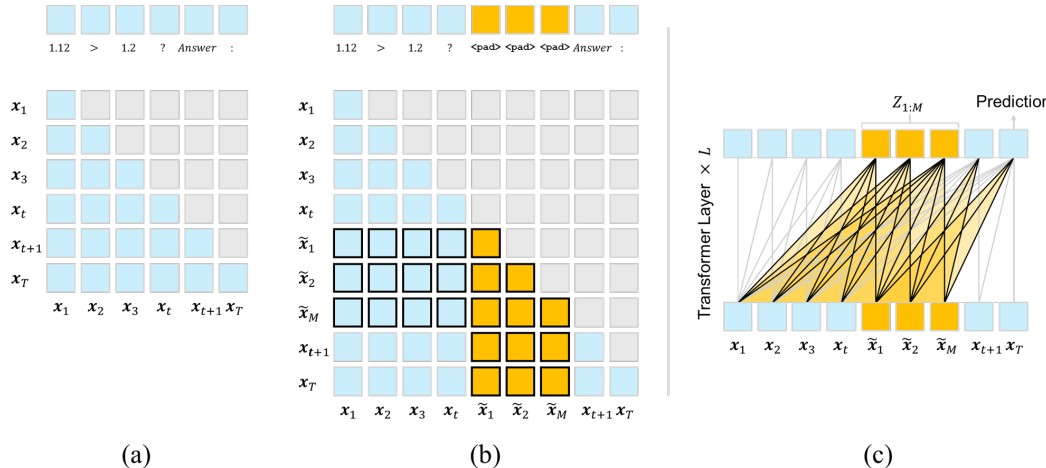

Figure 1: Attention patterns in a transformer decoder with causal masking. (a) The original prompt and its attention map. (b) The same prompt with filler tokens inserted within the prompt, and its attention map. Yellow boxes mark the extended *memories* induced by the fillers. (c) An equivalent edge-wise depiction of the attention mapping. The black bounding boxes in (b) and the black solid lines in (c) denote the additional attention *operations* induced by the filler tokens.

To answer the questions, we adopt six characters as filler tokens to extend the input space: ' ' (*space*), '\n' (*enter*), '\t' (*tab*), '.' (*period*), '<pad>' (*pad*), '-' (*dash*). We vary the number of filler tokens from 16 up to 8192 and place these tokens before and after the final 'Answer:' token of the input. In addition, we experiment with transformer-based causal decoder-only language models ranging from 1.7B to 32B parameters, and examine intermediate checkpoints to study when and how this expanded space becomes effective for problem-solving. To further investigate what actually occurs within this expanded space with the original inputs, we analyze attention maps.

Our experiments show that models can exploit filler tokens as expanded computation spaces even without explicit training. While the optimal token types and counts vary across models, some tokens consistently improve performance, with smaller models benefiting more, which is likely due to their limited computational capacity. Analysis of intermediate PT and IT checkpoints indicates that models progressively learn to leverage these spaces during training. Attention map analysis further reveals that the expanded spaces serve as meaningful extensions of the original attention mechanism, sometimes attending directly to questions or answer options and contributing to answer inference.

## 2 RELATED WORK

### 2.1 SLOW THINKING FOR IMPROVED REASONING

Dual-system theory (Daw et al., 2005), which posits that human cognition operates through two distinct modes, has also been invoked in prior machine learning research. System 1 processes problems quickly in an immediate and automatic manner, while System 2 is slower but more analytical, goes step by step, leading to more accurate and higher-level decision-making (Evans, 1984; Kahneman, 2003). Transformer-based language models have been known to be proficient at fast decision-making but have shown weaknesses in handling complex reasoning. However, recent models exhibit sophisticated reasoning capabilities, in some cases comparable to those of human experts (Hurst et al., 2024; Liu et al., 2024a; Team et al., 2023). Recent developments show that large language models (LLMs) are evolving from fast, intuitive System 1 processing toward slower but more deliberate System 2 reasoning (slow thinking) (Li et al., 2025; Weston & Sukhbaatar, 2023). LLMs can infer solutions to unseen problems by leveraging patterns from a few given samples or shots. In particular, they are developing sophisticated reasoning skills that mimic human abilities, such as decomposing problems step by step (e.g., chain-of-thought, CoT) and solving them in a structured, sequential manner (Hua & Zhang, 2022; Wei et al., 2022; Wang et al., 2022; Zelikman et al., 2024b).

## 2.2 TOKENS FOR THINKING

Previous studies have explored methodologies that incorporate dummy tokens or special tokens during the pre-training (PT) or fine-tuning (FT) stages, aiming to improve the reasoning of the models. First, in the DeepSeek-R1 (Guo et al., 2025) model, the tokens `<think>` and `</think>` are utilized to guide the model to perform a thinking process for answer inference within the enclosed span. Similarly, Zelikman et al. (2024a) proposed a method in which, at each token generation step, the tokens `<|startofthought|>` and `<|endofthought|>` were used to prompt the model to generate multiple rationales explaining the future text, thereby improving prediction. Herel & Mikolov (2024) demonstrated that, within a recurrent neural network (RNN) (Hochreiter & Schmidhuber, 1997) architecture, inserting thinking tokens `<T>` between input tokens reduces the perplexity of correct answers in complex mathematical computations. Additionally, ongoing research has explored the use of special thinking tokens to prompt models to engage in improved reasoning Fan et al. (2025); Yoon et al. (2025).

The works most closely aligned with our study are Pfau et al. (2024) and Goyal et al. (2023). Pfau et al. (2024) shows that transformers can use meaningless filler tokens in place of chain-of-thought for problem-solving when they are properly trained. It introduces a method for generating the synthetic data with filler tokens and training to converge the models. In Goyal et al. (2023), multiple `<PAUSE>` tokens are inserted into the model input to delay the answer, rather than producing it immediately. To this end, the input is deliberately modified during the PT and FT stages to include `<PAUSE>` tokens, making the model learn them. They report that delaying the responses using these special tokens leads to performance improvements across several benchmarks. In Lanham et al. (2023), filler tokens (e.g., '......') are replaced with chain-of-thought (CoT) sentences to study what contributes to the performance, but the performance decreases without CoT sentences, suggesting that training should be executed to use filler tokens for reasoning. Whereas prior works rely on training such tokens to elicit reasoning or thinking abilities from models, our approach explores the provision of additional extended space in the model input without training the tokens, instead utilizing tokens that the model has seen during training, to evaluate their effect.

## 3 EXPERIMENTS

### 3.1 NEXT-TOKEN PREDICTION

Causal decoder-only language models, consisting of a vocabulary $\mathcal{V}$ and $L$ transformer layers, are given an input $\boldsymbol{x}_{1:T} \in \mathcal{V}^T$ with $T$ length of tokens. For each layer $l \in [1, L]$, an intermediate vector $\mathbf{z}_t^{(l)}$, for each input token $\boldsymbol{x}_t$, is obtained. Based on the last token vector of the last layer $\mathbf{z}_T^{(L)}$, the models predict the most likely next token $\boldsymbol{x}_{T+1}$. In each transformer block, it takes a matrix of $T$ vectors of $D$ hidden state dimension size, $\boldsymbol{Z}_{1:T}^l = [\boldsymbol{z}_1, ..., \boldsymbol{z}_T] \in \mathbb{R}^{D \times T}$, as its input, and transform it into the output matrix $\boldsymbol{Z}_{1:T}^{l+1} \in \mathbb{R}^{D \times T}$, with its internal attention, feedforward and layer-norm modules. The attention module takes query vector $\boldsymbol{Q}_t \in \mathbb{R}^D$ for each input position $t$, and key and value matrices $\boldsymbol{K}_{1:t}, \boldsymbol{V}_{1:t} \in \mathbb{R}^{t \times D}$ of all previous positions. Then the attention output passes feedforward and layer-norm modules as follows:

$$
\begin{aligned}
\boldsymbol{a}_t^l &= LayerNorm(Attention(\boldsymbol{Q}_t, \boldsymbol{K}_{1:t}, \boldsymbol{V}_{1:t}) + \boldsymbol{z}_t^l) \\
\boldsymbol{z}_t^{l+1} &= LayerNorm(FeedForward(\boldsymbol{a}_t^l) + \boldsymbol{a}_t^l).
\end{aligned}
\tag{1}
$$

Note that, depending on the model, layer normalization can be applied either before or after the attention operation.

### 3.2 SCALING COMPUTATION SPACES

As illustrated in Figure 1 (c), given an input sequence $\boldsymbol{x}_{1:T}$, we insert filler tokens $\tilde{\boldsymbol{x}}$ of length $M$ (tokens in *yellow*), thereby extending the input to a sequence of length $T + M$. This extension produces an additional set of $M$ intermediate vectors, denoted as $\boldsymbol{Z}_{1:M}^l = [\boldsymbol{z}_1^l, \dots, \boldsymbol{z}_M^l]$ for each $l$-th transformer layer, within the model. The model then predicts the next token by jointly considering the original input and the inserted filler tokens. We hypothesize that these additional positions provide extra representational capacity beyond the original input length, functioning as *expanded computation spaces* (ECS) that support improved reasoning capabilities:

$$ECS_M = f(x_{1:T+M}) \setminus f(x_{1:T})$$
$$= \boldsymbol{Z}_{1:M} = [\boldsymbol{z}_1, \ldots, \boldsymbol{z}_M] \qquad (2)$$

Here, $f(\cdot)$ denotes the hidden representation function of the transformer-based decoder-only model given the input, and the operator '$\setminus$' denotes the set difference, meaning that $ECS$ corresponds to the *expanded computation spaces*, which are the additional computation spaces introduced by the inserted filler tokens. In terms of attention scores, for auto-regressive language modeling, non-lower-triangular scores in the attention map are masked out (grey regions in Figure 1 (a) and (b)) as it is not a bidirectional operation. With the original input of length $T$, attention computation involves $T^2$ scores, of which $T(T-1)/2$ are masked. When the input is augmented with $M$ filler tokens, the attention complexity increases to $(T+M)^2$, with $(T+M)(T+M-1)/2$ scores masked. Consequently, the regions in black lines correspond to $ECS$.

We examine whether $ECS$ merely constitute redundant additions in the attention computation of transformer-based models, or whether they can actively contribute to model inference, and if so, in what manner. We hypothesize that such synthetic modifications of the input provide additional computational capacity for reasoning, thereby enhancing performance. However, since this intervention deviates from the way models are typically trained, we assume that its effectiveness may vary across models and settings, and in some cases may deteriorate the performance without additional training.

To investigate this, we first conduct experiments on question answering tasks (§ 4.1) and mathematics tasks (§ 4.2) with different models to identify the conditions under which expanded spaces are effective. We hypothesize that there exist specific models and tasks for which the effect is particularly pronounced. We analyze how the number and type of inserted tokens (§ 4.1.2, § 4.1.1), as well as their position within the input (§ 4.2.2), influence model performance. We further conjecture that smaller models may benefit more from such extensions, as their limited parameters could gain from additional horizontal computational capacity. Moreover, we expect the intervention to be more effective when the inserted tokens resemble patterns frequently observed during the training phase (e.g., sequences of ' ' (space) tokens are likely more common than repeated '%', which may primarily arise from the original text sources). Finally, we investigate at what stage of training such capabilities emerge (§ 4.3). To this end, we experiment with the pretrained (PT) and instruction-tuned (IT) models within the same model family and evaluate performance across intermediate checkpoints. This allows us to trace the development of expanded spaces throughout the training process. Finally, we analyze attention maps of inputs with expanded computation spaces to investigate the computations taking place within them (§ 4.4).

### 3.3 EXPERIMENTAL SETTINGS

#### 3.3.1 MODELS

We experiment with models with the number of parameters from 1.7B to 32B that are publicly available on HuggingFace's transformers libraries (Wolf et al., 2020). We adopt HuggingFace's `SmolLM2-1.7B-Instruct` model, which is a 1.7B-sized model. For the 4B size model, we use Google Gemma team's `Gemma-3-4B-it` (Team et al., 2025), which has 1024 context length. For the 8B model, we adopt Meta's `Llama-3.1-8B-Instruct` (Grattafiori et al., 2024) with 131072 context length. For 14B model, we adopt Qwen team's `Qwen2.5-14B-IT` (Qwen et al., 2025) (context length of 131072), AllenAI's OLMo-2 (OLMo et al., 2025) (context length of 4096). OLMo-2 has released its intermediate checkpoints for pretraining (`OLMo-2-1124-13B`) and instruction-tuning (`OLMo-2-1124-13B-Instruct`), so we exploit these checkpoints for tracking with the training steps. For the biggest size model in our experiments, we use the `Qwen2.5-32B-Instruct` model.

#### 3.3.2 DATASETS

We primarily experiment with two datasets: Measuring Massive Multitask Language Understanding (MMLU, Hendrycks et al. (2021)) and AI2 Reasoning Challenge (ARC, Clark et al. (2018)) for measuring the model's ability in general domain question answering. MMLU is composed of 57 tasks, a total of 14079 samples that can be categorized into (1) Humanities, (2) Social Science, (3) Science, Technology, Engineering, and Mathematics (STEM), and (4) Other. We adopt ARC

*challenge* test subset, which has 1172 samples in science questions with mostly 4 options, but also with 3 and 5 options. We additionally consider the mathematics dataset GSM8K and MATH-500 [1] to measure the deeper reasoning ability of models. GSM8K has 1819 test set examples of school mathematics. MATH-500 consists of 500 problems from the MATH benchmark that OpenAI created in their paper (Lightman et al., 2023)

### 3.3.3 EVALUATION

We evaluate model performance using the last token logits of the final layer. For the MMLU and ARC datasets, which consist of multiple-choice questions with mostly four options, we provide the model with the input sequence: `chat template` (for IT models), `task instruction`, `context` (if exists), `question`, `answer options`, and the prompt '`Answer:`'. From the final logits, we extract only those corresponding to the option tokens (A–D), apply a softmax over them, and select the option with the highest probability as the predicted answer. For the math tasks, the model is required to generate an answer in free-form text, and the final prediction is extracted from the output using a predefined rule to match the exact value, following the existing implementation.[2]

For our experiments, we use a zero-shot setting, providing the models only with the dataset samples without any additional task-related examples, and we report the average result of three runs with different seeds. The inference time for each experiment varies from 3 minutes to over 3 hours on one H100 GPU, according to the model size, input length, and batch size.

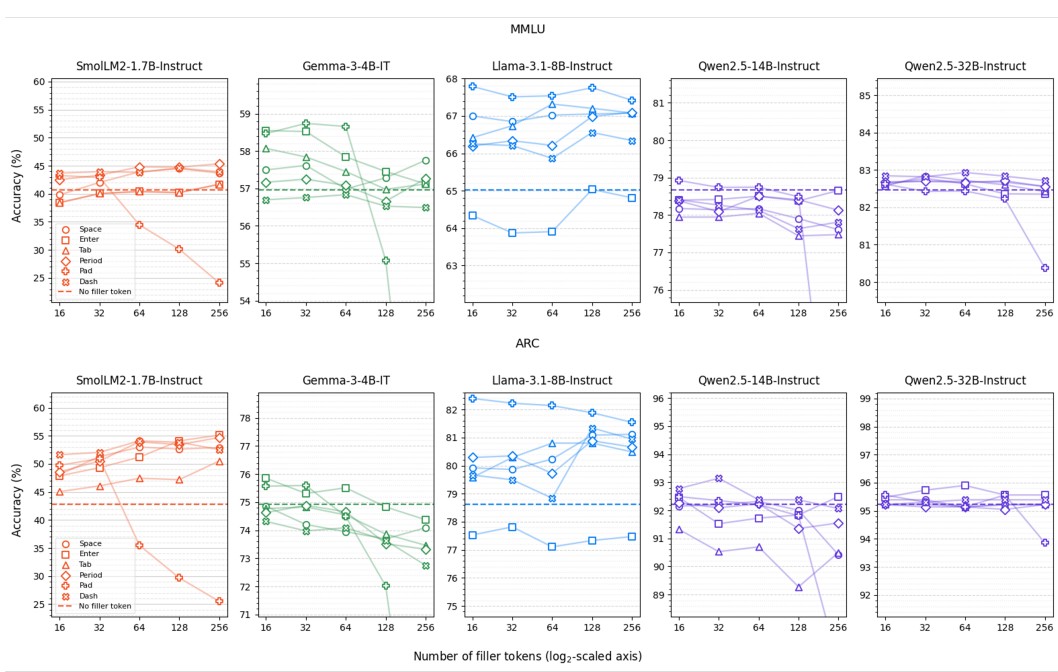

Figure 2: MMLU and ARC accuracy (%) results of four different models. Each model exhibits different token types and quantities that are most effective. Smaller models tend to benefit from the additional space provided by almost any tokens, using them to improve performance. In contrast, models with sufficient capacity for solving the task gain relatively less from such expansions.

---

[1] https://huggingface.co/datasets/HuggingFaceH4/MATH-500
[2] https://github.com/EleutherAI/lm-evaluation-harness/blob/main/lm_eval/tasks/hendrycks_math/utils.py

## 4 RESULTS

### 4.1 EFFECT ON QUESTION ANSWERING TASKS

We first investigate the effect of adding filler tokens by evaluating the multiple-choice question answering performance of models on MMLU, a general-domain QA benchmark, and ARC, a science-domain QA task. Through these experiments, we identify which types of filler tokens, and in what quantities, can positively influence model performance. We present the full results in Appendix A.2 and A.3.

#### 4.1.1 TOKEN TYPES

In our experiments, we consider six types of tokens: ' ' (*space*), '\n' (*enter*), '\t' (*tab*), '.' (*period*), '<pad>' (*pad*), '-' (*dash*). Across models, we observe accuracy improvements of up to 12.372 percentage points, as presented in Figure 2. For the SmolLM model, the period token yields the best overall performance on MMLU, while the enter token is most effective on ARC. In contrast, the <pad> token severely degrades performance when more than 64 tokens are inserted for all models except Llama-3.1-8B-IT. Interestingly, Llama achieves the highest performance with the <pad> token, which can be attributed to the fact that it does not employ a dedicated <pad> token but instead uses the <eos> token in its place. In this case, even with 256 additional tokens, performance remains substantially higher than the baseline. Qwen2.5-14B-IT model shows far smaller gains compared to other models. For Qwen2.5-32B-IT model, unlike the Qwen2.5-14B-IT, we observe no early degradation of performance when additional tokens are introduced. Instead, the expanded space contributes to performance improvements. Although the magnitude of improvement is substantially smaller than that of smaller models, the gains are steady and consistent.

#### 4.1.2 TOKEN NUMBERS

Overall, we observe that performance tends to deteriorate as more tokens are added, with a sharp decline occurring for the <pad> token once the number exceeds 64. We attribute this behavior to the characteristics of the training corpus and preprocessing, where excessive repetitions of filler-like tokens are relatively rare. Since the models have not been trained on such inputs, an overly large number of added tokens cannot be effectively utilized and, instead, appear to hinder performance, resembling the lost-in-the-middle phenomenon (Liu et al., 2024b; Wright et al., 2025). It was observed that once the number of added tokens surpassed 1024, the accuracy consistently declined to the 20% range across almost all cases, as shown in Table 1 in Appendix A.1.

While all models show performance degradation beyond a certain input length, the SmolLM2-1.7B-IT model exhibits the opposite trend, with accuracy continuing to improve as the number of tokens increases up to 256. This suggests that, in SmolLM, the additional tokens are effectively utilized for reasoning, compensating for its limited parameter capacity by exploiting the synthetically expanded computation space. In contrast, larger models—already equipped with sufficient parameters to achieve strong performance—benefit relatively less from such extensions.

### 4.2 EFFECT ON MATHEMATICS TASKS

To evaluate the mathematical reasoning ability of the models, we assess their generation performance on math tasks. Figure 3 reports results on MATH-500 and GSM8K. The charts show the models' ability to generate correct answers when given CoT rationales included in the dataset.

#### 4.2.1 MATH REASONING

The result trends differ slightly between the two benchmarks as in Figure 3 (a). Similar to the QA experiments, the smallest model exhibits the most pronounced performance gains. For Gemma-3-4B-IT, performance peaks when 256 tokens are added in MATH-500, suggesting once again that the expanded space compensates for the limitations imposed by its smaller parameter size. We also observe that as the number of inserted tokens increases, performance suddenly drops beyond a certain point. In MATH-500, Llama-3.1-8B-IT initially exhibits a decline in performance, but its accuracy improves markedly when 256 filler tokens are added. In contrast, on GSM8K, Llama shows

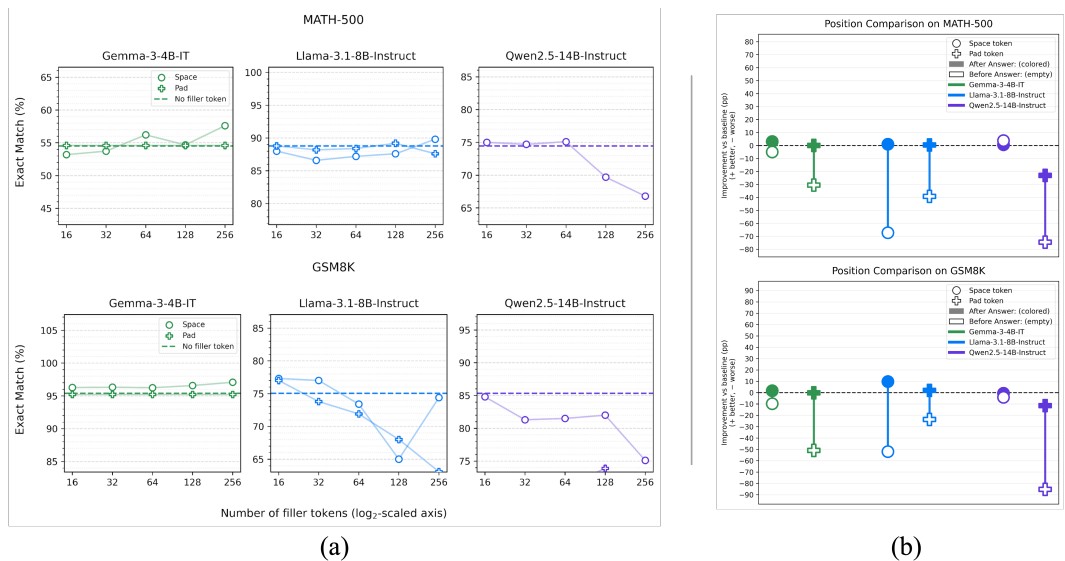

(a)                                                (b)

Figure 3: (a) MATH-500 and GSM8K exact match (%) scores. Models perform better with ' ' tokens than with `<pad>` tokens. (b) Position comparison between the filler tokens placed before and after the 'Answer:' token. When the filler tokens are placed *after* 'Answer:' token, the performances significantly degrade.

an initial performance gain, but its accuracy gradually decreases as more tokens are introduced. Qwen2.5-14B-IT, on the other hand, does not appear to benefit from the expanded space in either task: the space token yields only a minor improvement in the very early stages, while the `<pad>` token provides no observable effect.

### 4.2.2 INSERTION POSITION

We investigate two token types in this experiment: a ' ' token that generally yielded strong performance in prior experiments and `<pad>` that did not contribute to performance. Instead, we focus on specific cases: inserting tokens immediately before 'Answer:' token versus inserting them after 'Answer:' token. As shown in Figure 3 (b), placing filler tokens before the `Answer:` token (colored) leads to better performance compared to placing filler tokens after it (empty). Even if a large number of filler tokens are present, the model can still generate an appropriate answer when `Answer:` appears at the very end, since the final input naturally prompts the next-token prediction to produce an answer. However, if many filler tokens follow `Answer:`, the model is more likely to predict another filler token as the next token, which provides little benefit for answer generation. Therefore, when a large amount of artificial tokens is added to the input, positioning the answer-prompting token at the very end of the sequence appears crucial for effectively utilizing the additional computational space.

### 4.3 DEVELOPMENT OF EXPANDED COMPUTATION SPACE

We investigate when models begin to exploit filler tokens as expanded computational spaces by conducting experiments across the full training trajectory of OLMo-2-1124-13B on the ARC task, using publicly available checkpoints from the initial pretraining (PT) through instruction-tuning (IT) stages. PT is divided into Stage 1, comprising over 90% of training, and Stage 2, accounting for roughly 5–10%. Stage 1 checkpoints of the released OLMo-2 PT model were saved every 1K steps, totaling 590K steps (∼5T tokens), and we experiment on checkpoints corresponding to every 200B tokens. For Stage 2, training weights were constructed from three ingredients; we select ingredient 4 and use checkpoints saved every 1K steps, covering ∼35K steps (∼300B tokens). As shown in Figure 4, adding 64 *enter* tokens initially lowers performance compared to the original input. Only from mid to late Stage 2 does performance approach and slightly exceed the baseline, suggesting

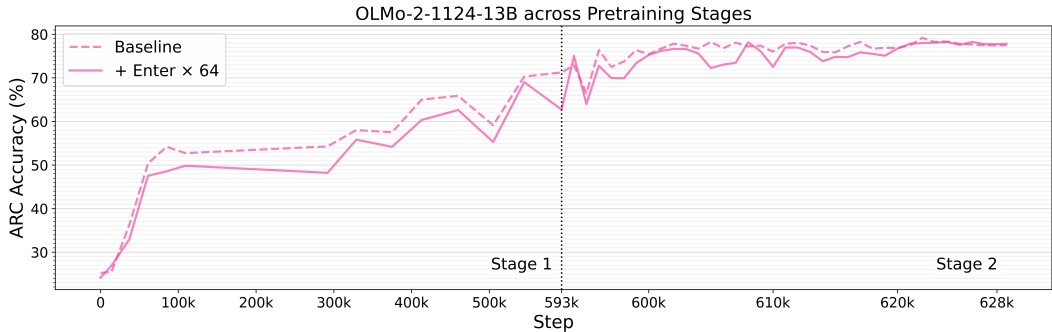

Figure 4: ARC results across PT checkpoints of OLMo-2-1124-13B. PT Stage 1 is compressed and Stage 2 is expanded for readability.

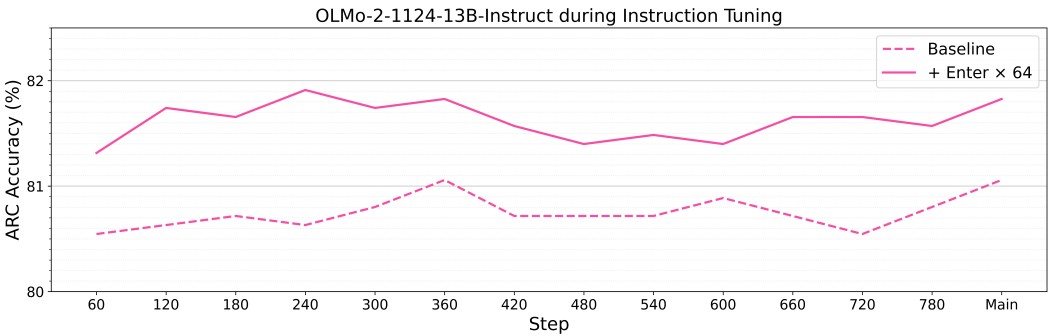

Figure 5: ARC results across IT checkpoints of OLMo-2-1124-13B-Instruct.

that the model needs sufficient exposure to diverse data and a certain level of language proficiency before it can effectively leverage filler tokens as additional computational space.

For the OLMo-2 IT model, the instruction-tuning phase spans from step 60 to the final step, with 14 checkpoints sampled at intervals of 60 steps. The results are shown in Figure 5. The model exhibits only minor variations at score levels above 80.5, achieving its highest performance at the main checkpoint. When filler tokens are added, the model follows a similar overall performance trend but maintains approximately 1% point higher scores. After PT, in IT stage, performance improves consistently across all checkpoints, suggesting that the model can leverage filler tokens as additional computational space once it has sufficient prior knowledge of them.

## 4.4 INTERPRETING INNER WORKINGS

To see the relationship between the original input and the extended spaces, we look into the attention map of the input extended with filler tokens, as in Figure 6. We analyze an example from the Gemma-3-4B-IT, where the model fails to predict the correct answer with only the original input but succeeds when filler tokens are added. For visualization, we omit the chat template and task explanation for clarity.

As shown in (a) in Figure 6, in some cases, the attention scores within the added spaces are higher in average than in other parts, particularly in the early layers. In (b), the entire input shows strong attention to the question (upper white regions), while the filler tokens attend heavily to the  token and the first filler token (red regions). Notably, the filler tokens consistently attend to the original input in a uniform manner, a phenomenon frequently observed across many layers and heads. This suggests that filler tokens contribute relatively evenly, regardless of their position. As in (c), certain filler tokens exhibit strong attention to specific parts of the question, such as the word 'kinase' (lower-left dark red point), as well as to tokens within the answer options. This indicates

[Sample #123 of Clinical Knowledge in MMLU]

The creatine kinase reaction is:

   A. irreversible   B. slow compared with glycolysis   C. not activated until all ATP has been used up   **D. inhibited by low ph in the muscle**

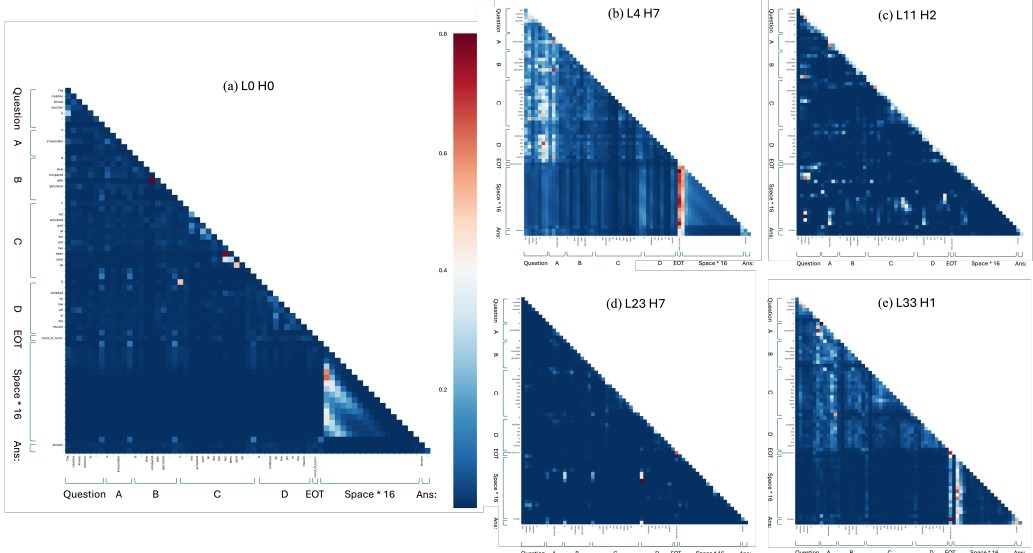

Figure 6: Attention maps of `Gemma-3-4B-IT` model with 16 of ' ' tokens. This is the sample #123 of clinical knowledge in MMLU dataset. For simplicity, we do not include the texts in chat templates (e.g., 'You are a helpful assistant...') and the task explanations (e.g., 'This is a multiple-choice...') in this example. 'L' and 'H' in the figures indicate *layer* and *head* index, respectively.

that, in the middle layers, filler tokens participate in interpreting both the question and the options. In (d), at the 23rd layer out of 34 layers, the filler token space shows strong attention to option D, which corresponds to the correct answer. This suggests that the decision toward the correct answer begins to emerge in the mid-to-late layers. As in (e), in the final layer, the filler tokens give the highest attention to the `` and answer tokens, indicating preparation for generating the final output.

These examples demonstrate that filler tokens do not merely serve as meaningless extensions of the input space, but rather attend to important information in the question and options, and influence the process of answer selection. Additional examples from other models and samples are provided in Appendix A.4.

## 5  CONCLUSION

In this work, we introduced an intriguing phenomenon where the insertion of filler tokens at inference time leads to performance improvements in language models. Across models of varying sizes, we observed that although the effective token types and quantities differ, the presence of expanded spaces contributes to better performance on both QA and mathematical reasoning tasks. Through experiments with intermediate PT and IT checkpoints, we further demonstrated that models acquire the ability to utilize these additional spaces during the pretraining phase. Moreover, smaller models benefit more substantially, suggesting that the expanded space compensates for their limited parameter capacity. Finally, our attention map analysis revealed that these spaces are not simply redundant extensions of the input, but rather serve as spaces where meaningful computations for answer inference take place.

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

# A APPENDIX

## A.1 RESULTS WITH LONGER FILLER TOKENS

Table 1: MMLU and ARC accuracy (%) with ' ' (space) tokens added. The numbers highlighted in yellow indicate performance improvements compared to the case without filler tokens. Smaller models exhibit larger performance gains, while performance begins to break down once more than 1024 tokens are added.

| | MMLU | | | | |
|---|---|---|---|---|---|
| $M$ | SmolLM2-1.7B-IT | Gemma-3-4b-it | Llama-3.1-8B-IT | Qwen2.5-14B-IT | Qwen2.5-32B-IT |
| 0 | 40.639 | 59.953 | 65.012 | 78.663 | 82.457 |
| 16 | 39.805 | 57.496 | 65.999 | 78.173 | 82.657 |
| 32 | 42.023 | 57.617 | 66.846 | 78.141 | 82.708 |
| 64 | 43.920 | 56.986 | 67.021 | 78.168 | 82.685 |
| 128 | 44.505 | 57.284 | 67.050 | 77.903 | 82.673 |
| 256 | 43.691 | 57.756 | 67.076 | 77.608 | 82.556 |
| 512 | - | - | 67.546 | 77.599 | 82.449 |
| 1024 | - | - | 64.394 | 78.662 | 81.969 |
| 2048 | - | - | 32.517 | 78.178 | 45.889 |
| 4096 | - | - | 27.453 | 48.695 | 27.328 |
| 8192 | - | - | 25.739 | 25.062 | 17.242 |

| | ARC | | | | |
|---|---|---|---|---|---|
| $M$ | SmolLM2-1.7B-IT | Gemma-3-4b-it | Llama-3.1-8B-IT | Qwen2.5-14B-IT | Qwen2.5-32B-IT |
| 0 | 42.747 | 74.915 | 78.584 | 92.150 | 95.222 |
| 16 | 48.123 | 75.171 | 79.949 | 92.150 | 95.392 |
| 32 | 51.280 | 74.232 | 79.778 | 92.235 | 95.392 |
| 64 | 52.986 | 73.891 | 80.290 | 92.235 | 95.137 |
| 128 | 52.645 | 73.635 | 81.058 | 91.980 | 95.222 |
| 256 | 52.901 | 73.976 | 80.973 | 90.358 | 95.222 |
| 512 | - | - | 81.058 | 90.700 | 95.051 |
| 1024 | - | - | 71.843 | 91.894 | 94.795 |
| 2048 | - | - | 25.683 | 92.235 | 36.775 |
| 4096 | - | - | 24.573 | 61.775 | 23.038 |
| 8192 | - | - | 26.365 | 22.611 | 14.420 |

## A.2 MMLU Results

Figure 7: MMLU accuracy scores of models with each filler token type.

## A.3 ARC RESULTS

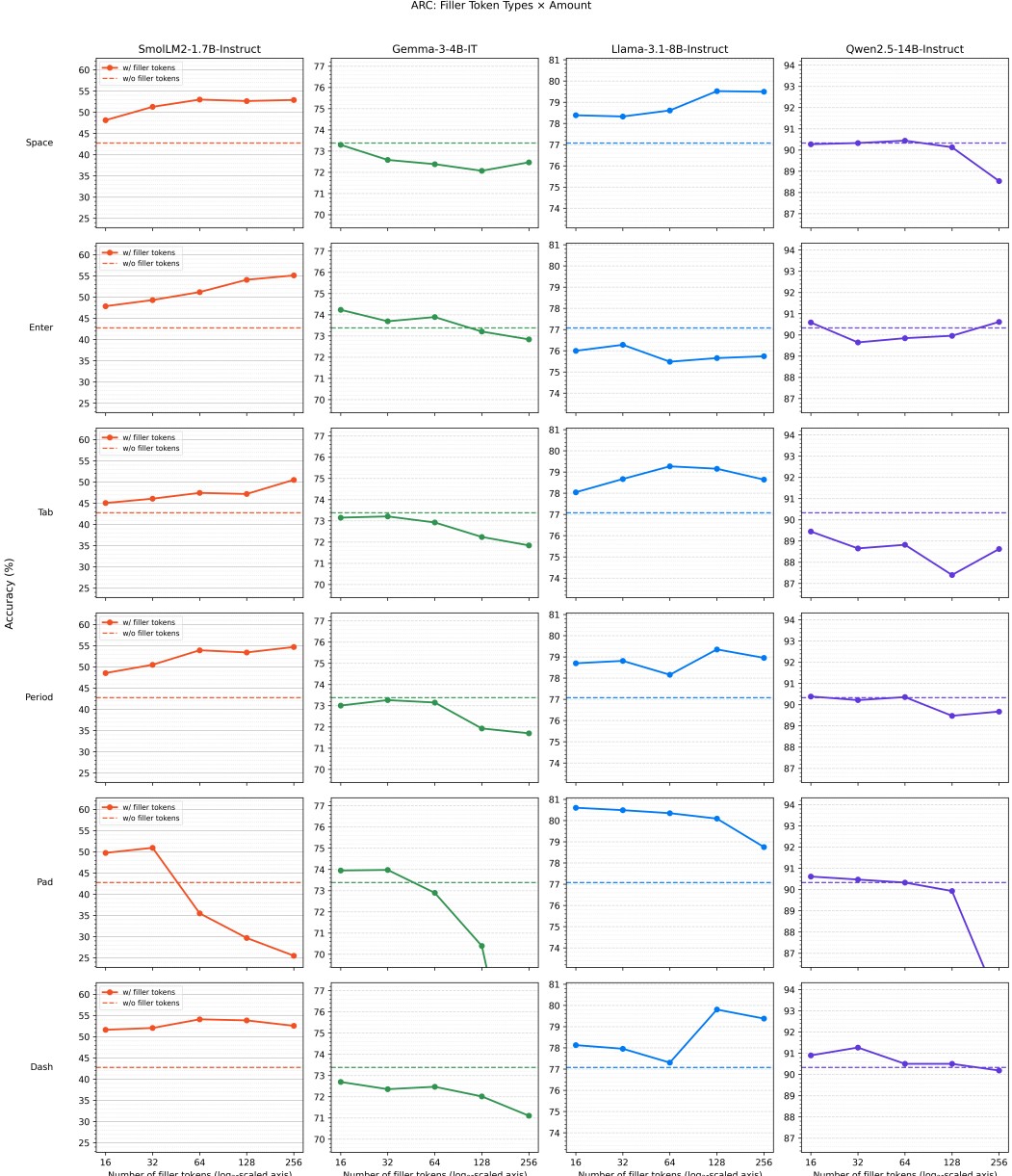

Figure 8: ARC accuracy scores of models with each filler token type.

## A.4 ATTENTION MAP EXAMPLES

Figure 9: SmolLM2-1.7B-Instruct's attention maps with 64 of period tokens. This sample is 15 of elementary mathematics in MMLU.

