# OpenReview forum: "Expanding Computation Spaces of LLMs at Inference Time"
_ICLR.cc/2026/Conference — Submitted to ICLR 2026_

### Official Review · Reviewer_3eZp · 2025-10-24

**Soundness:** 2
**Presentation:** 3
**Contribution:** 1
**Rating:** 2
**Confidence:** 4

**Summary:**

This paper investigates whether large language models can leverage artificially inserted filler tokens—without any additional training—to enhance reasoning performance. Building on the intuition that chain-of-thought (CoT) prompting benefits not only from explicit reasoning steps but also from the larger “computational space” of longer inputs, the authors systematically study what kinds, amounts, and insertion locations of filler tokens are effective. Through experiments across multiple open-domain QA and math benchmarks using models from 1.7B to 32B parameters, they find that inserting familiar filler characters (e.g., spaces, newlines, dashes) directly before the final Answer: token can significantly improve accuracy, especially in smaller models (up to +12.37 percentage points). Attention-map analyses reveal that these expanded spaces participate in meaningful computation, continuing or redirecting attention toward relevant parts of the problem.

**Strengths:**

1. The experiments cover two categories of tasks—QA (MMLU, ARC) and mathematics (GSM8K, MATH-500)—using a range of instruction-tuned models from small to large (including Qwen2.5-32B-Instruct). Results are reported in a zero-shot setting averaged over three random seeds. For multiple-choice questions, the answer is extracted by applying a softmax over the final-layer logits of options A–D, and for math problems, the numerical answer is extracted following existing evaluation rules. The evaluation interface is fully transparent.

2. A comprehensive factor analysis is conducted: systematic ablations are performed on token type (space, newline, tab, period, <pad>, dash), quantity, and insertion position, revealing several consistent experimental patterns.

3. A preliminary set of actionable guidelines is provided, summarizing multiple practical insights with direct reference value for deployment.

**Weaknesses:**

1. I am confused and skeptical about the motivation of the paper. If the goal is to alleviate the time consumption caused by the lack of parallelism in CoT reasoning, then according to the experimental results reported in the paper, this method seems to save only about the generation time of 256 tokens, which appears to bring minimal benefit for long-text reasoning. If the goal is to propose an alternative to CoT, the paper lacks direct comparative experiments on the answer quality obtained by CoT.

2. The authors should add a “same token budget” comparison with CoT: the paper does not take CoT prompts with the same number of tokens as a baseline for direct comparison, making it difficult to answer the key question of whether ECS performs better (or worse) than short CoT under the same token or time cost. The current results mostly verify that “merely adding positions can bring gains,” but if ECS does not show clear advantages over CoT in some aspects, its practical value will be greatly discounted.

3. The paper has not yet summarized general, actionable insights that are independent of specific tasks.

**Questions:**

Could the decline in accuracy with an increasing number of tokens in the ECS be related to the inherent difficulty of the problem itself? Perhaps the additional tokens cause the model to overthink, leading it to second-guess its own reasoning.

---

> ### Author Response · Authors · 2025-11-20
>
> Thank you for taking the time to review our paper. Please find our answers below:
>
> > I am confused and skeptical about the motivation of the paper. If the goal is to alleviate the time consumption caused by the lack of parallelism in CoT reasoning, then according to the experimental results reported in the paper, this method seems to save only about the generation time of 256 tokens, which appears to bring minimal benefit for long-text reasoning. If the goal is to propose an alternative to CoT, the paper lacks direct comparative experiments on the answer quality obtained by CoT.
>
> > The authors should add a “same token budget” comparison with CoT: the paper does not take CoT prompts with the same number of tokens as a baseline for direct comparison, making it difficult to answer the key question of whether ECS performs better (or worse) than short CoT under the same token or time cost. The current results mostly verify that “merely adding positions can bring gains,” but if ECS does not show clear advantages over CoT in some aspects, its practical value will be greatly discounted.
>
> The study of replacing filler tokens with CoT prompts has already been conducted in [1]. However, in that study, the model had never been trained on specifically transformed filler tokens, so they created additional synthetic datasets for training and reported performance comparable to CoT on certain tasks. Inspired by this, we started our research with the idea that additional input space itself might be utilized by the model for reasoning.
>
> However, without such training, adding a reasoning trajectory (CoT) input to the model naturally contains more information than seemingly meaningless filler tokens, so it inevitably yields better performance; thus, we did not treat them as directly comparable. In fact, the results in Figure 3 include CoT prompt inputs provided by the datasets for both Math500 and GSM8K. When CoT inputs were entirely removed, the EM scores dropped below roughly 10%, highlighting the large performance gap between having CoT and not having it. Nevertheless, whether CoT was included or not, we observed performance improvements when filler tokens were additionally provided. This shows that while it is unreasonable to replace CoT entirely with filler tokens if the model has not learned the filler token format, filler tokens can still contribute to performance improvements when used in addition to CoT.
>
> [1] Let's Think Dot by Dot: Hidden Computation in Transformer Language Models

---

> ### Author Response · Authors · 2025-11-20
>
> > The paper has not yet summarized general, actionable insights that are independent of specific tasks.
>
> Our study did not aim to propose a method to improve reasoning. Rather, we investigated and experimentally tested whether LLMs can utilize seemingly meaningless filler tokens given in a form they had not encountered during PT/IT. The results showed that this ability varies depending on the model type, size, and the tokens provided. However, it was clear from the attention maps that models can leverage the expanded input space for reasoning. The attention maps revealed that the additional tokens focused on critical input tokens, answer candidates such as A–D, and functional tokens like EOT, contributing to the model’s inference of the correct answer.

---

> ### Author Response · Authors · 2025-11-20
>
> > Could the decline in accuracy with an increasing number of tokens in the ECS be related to the inherent difficulty of the problem itself? Perhaps the additional tokens cause the model to overthink, leading it to second-guess its own reasoning.
>
> We experimented not with completely random token sequences, but with repeated spaces, line breaks, periods, and similar tokens. This is because our study aims to extend the input artificially during inference without the model having learned such patterns. Based on previous studies showing that a model trained on a certain amount of completely random tokens can achieve performance comparable to CoT, we can infer that performance drops occur because the model is unfamiliar with this type of dataset. Interestingly, however, we found that some models possess a latent ability to utilize these tokens even without prior training, and that increasing the input space for smaller models can enhance their performance. Additionally, the performance drop observed with longer inputs does not only apply to the additional tokens but also seems to stem from the “lost-in-the-middle” problem common in transformer-based models.
>
> As reviewer N9o6 noted, this is a somewhat expected result, and we will add an explanation of this in the paper.

---

> > ### Comment · Reviewer_3eZp · 2025-11-21
> >
> > Thank you for your reply and for clarifying the questions we raised. Based on your response, I think I can tentatively understand your paper as aiming to reveal certain phenomena, rather than to propose an algorithm that can be directly used in practice at this moment. If the phenomena are sufficiently “solid” and one can reasonably foresee their potential future value, then I would consider this an excellent piece of research (even if there is currently no new algorithm).
> >
> > However, the problem now is that I cannot imagine what kind of potential value such a finding would ultimately bring. It seems that any algorithms it might give rise to in the future could all be replaced by CoT, or would run into token capacity limitations (as you mentioned in the paper, where performance drops significantly after adding 256 padding tokens).
> >
> > As for the differences between PT/IT that you describe in the paper, I believe they merely reflect, from another angle, a common consensus in the field, namely that in the IT stage the model acquires instruction-following and reasoning abilities far beyond those obtained in the PT stage.
> >
> > Moreover, and most importantly, I believe that attributing “inconsistency of effects” to “model family/type” is a fundamental mistake. It is like asking a large model a question: different types of models will give different answers, and models from the same family (because they share part of the training data) will give similar answers. This is something we would naturally expect. In other words, I could also study various variants such as “repeating each decoded token n times during inference,” “inserting random symbols after each generated token during inference,” or “inserting random symbols after each token in the prefill stage,” and so on. But if no consistent conclusion is obtained, then all such work can only be regarded as experimental reports, and at the very least are not suitable for publication at a top conference like ICLR.
> >
> > We once again thank the reviewers for replying to each reviewer’s comments in such detail, but at the same time I also hope the authors will seriously reflect on why all four reviewers gave consistent evaluations of the initial submission. Please acknowledge the parts of the paper that were not done sufficiently well. Since it is possible to upload a new manuscript during the rebuttal period, please mark the new changes you make in colored text.
> >
> > If you feel that any of the above points diverge from your understanding of my opinion, I would be very happy to discuss them with you further.

---

### Official Review · Reviewer_6CYq · 2025-10-26

**Soundness:** 1
**Presentation:** 2
**Contribution:** 2
**Rating:** 2
**Confidence:** 4

**Summary:**

This paper investigates the impact of adding filler tokens across types, counts, and positions. It also examines when during training this phenomenon emerges. Attention-map analysis shows that these tokens focus on relevant parts of the input and help the model arrive at the answer.

**Strengths:**

- The paper examines the impact of expanding computation space by adding filler tokens at inference time.

- It reports observations across several factors, including model type and size, as well as token types and counts.

**Weaknesses:**

The overall patterns of results and explanations are not consistent across model types and sizes, as well as token types and counts.

- The explanations are sometimes inconsistent. For example, for Qwen2.5-14B vs. 32B, the paper argues that larger models benefit from filler tokens, yet for SmolLM (1.7B) it claims the additional tokens are “effectively utilized for reasoning,” suggesting a different rationale for small models. In line with this concern, more experiments are needed to substantiate the claims.

- The paper argues that additional tokens compensate for limited parameters in small models, but it evaluates only SmolLM as the “small model.” More small models should be tested to support generalization.

- Several reported patterns do not generalize well. For instance, the paper claims that adding more than 64 <pad> tokens degrades performance, attributing this to its rarity during training. However, repetition of other tokens may also be rare in typical training data.

- According to Llama’s explanation of <pad> vs. <eos>, the training dataset influences these patterns. However, the impact of filler tokens in Qwen2.5-14B and 32B does not align, even though they are from the same model family.

- The checkpoint analysis shows no clear trends. Pretraining (PT) checkpoints do not improve with filler tokens, while instruction-tuning (IT) checkpoints likewise fail to show a consistent pattern of improvement across the training process.

- The attention map analysis hardly explains why the improvements occurred. Strong attention alone does not necessarily translate into better performance. Even so, the paper should present counterexamples (e.g., cases where the model answers incorrectly) and, where possible, quantify these patterns.


Typos:
- Line 119: The citation(s) should be in parentheses “( … )”.
- Equation (2): Should Z_{1:M} be Z_{T:M}?

**Questions:**

Please see the Weaknesses section.

---

> ### Author Response · Authors · 2025-11-20
>
> We truly appreciate the reviewer for the review and suggestions. Please find our answers below:
>
> >The overall patterns of results and explanations are not consistent across model types and sizes, as well as token types and counts.
>
> >Several reported patterns do not generalize well. For instance, the paper claims that adding more than 64 [object Object] tokens degrades performance, attributing this to its rarity during training. However, repetition of other tokens may also be rare in typical training data.
>
> According to our experimental results, larger models achieve higher baseline performance (without filler tokens) because they can fully utilize their sufficiently large representational space. The reason SmolLM shows the most pronounced performance change is that its representational space is limited; due to the lack of space to solve the task, providing filler tokens results in a sharp performance boost. However, our input manipulation represents an unusual type of input variation that was not considered during the PT/IT stages, so the ability to exploit it varies significantly across model types and sizes. As previous studies have shown, where training with filler tokens can yield performance comparable to CoT, we expect that additional training on the input format we introduce would lead to even greater improvements. Nevertheless, the scope of this study is to demonstrate that models possess the ability to utilize such input patterns during inference, even when they have never encountered them during training.

---

> ### Author Response · Authors · 2025-11-20
>
> > The explanations are sometimes inconsistent. For example, for Qwen2.5-14B vs. 32B, the paper argues that larger models benefit from filler tokens, yet for SmolLM (1.7B) it claims the additional tokens are “effectively utilized for reasoning,” suggesting a different rationale for small models. In line with this concern, more experiments are needed to substantiate the claims.
>
> We didn't argue that larger models gain more than smaller models from expanded spaces; rather, Qwen 14B model's gain is smaller than other models (including 32B models) (line 289-290). In conclusion, we found that this additional space was more effective in improving performance for smaller models with much fewer parameters.

---

> > ### Comment · Reviewer_6CYq · 2025-11-21
> >
> > "We didn't argue that larger models gain more than smaller models from expanded spaces; rather, Qwen 14B model's gain is smaller than other models (including 32B models) (line 289-290). In conclusion, we found that this additional space was more effective in improving performance for smaller models with much fewer parameters."
> >
> > I mentioned “for Qwen2.5-14B vs 32B.”

---

> > > ### Comment · Reviewer_6CYq · 2025-11-21
> > >
> > > Thank you authors for your response. However, most of your answers don’t fully address my concerns, so I’ll keep my original score.

---

> ### Author Response · Authors · 2025-11-20
>
> > The paper argues that additional tokens compensate for limited parameters in small models, but it evaluates only SmolLM as the “small model.” More small models should be tested to support generalization.
>
> Although we could not experiment with all types of models, we ran additional experiments below. As a result, we observed that for these models, performance consistently improves as the number of tokens increases. This confirms that, for small-sized models, expanding the input space clearly provides a benefit.
>
> | +FillerToken | Num |    White   |    Enter   |     Tab    | Dot        | Pad        | Dash       |
> |:------------:|:---:|:----------:|:----------:|:----------:|------------|------------|------------|
> |       X      |  -  |   48.976   |   48.976   |   48.976   |   48.976   |   48.976   |   48.976   |
> |       O      |  16 |   48.805   | **50.597** |   47.355   | **49.573** | **52.304** | **49.829** |
> |       O      |  32 | **49.147** | **51.024** | **49.915** | **49.744** | **53.840** | **50.171** |
> |       O      |  64 | **49.061** | **51.109** | **49.915** | **50.597** | **54.181** | **50.683** |
> |       O      | 128 |   48.720   | **50.768** |   43.601   | **50.000** | **54.181** | **50.085** |
> |       O      | 256 | **50.683** | **49.147** |   46.331   | **49.573** | **52.133** | **50.853** |
>
> The table above is the result of meta-llama/Llama-3.2-1B-Instruct on ARC dataset. Bold indicates improved performance than when the filler tokens are not added. In most cases, Llama 3.2 1B model with filler tokens show higher scores where the proportion of the improved scores are higher than other models.

---

> ### Author Response · Authors · 2025-11-20
>
> > According to Llama’s explanation of [object Object] vs. [object Object], the training dataset influences these patterns. However, the impact of filler tokens in Qwen2.5-14B and 32B does not align, even though they are from the same model family.
>
> We can’t see what kind of tokens you wrote. They appear as [object Object] vs. [object Object]. Could you please explain in words what kind of tokens they are?

---

> ### Author Response · Authors · 2025-11-20
>
> > The checkpoint analysis shows no clear trends. Pretraining (PT) checkpoints do not improve with filler tokens, while instruction-tuning (IT) checkpoints likewise fail to show a consistent pattern of improvement across the training process.
>
> In our paper, we aimed to investigate how model performance changes when filler tokens are added, and there was no training on the filler tokens themselves. Therefore, the experiments in Section 4.3 were not about training the model with filler tokens, but about testing the hundreds of mid-checkpoints provided by the OLMo team, spanning from the PT to IT stages, to see at which point the effect of filler tokens begins to appear. We were curious about when the performance-boosting effect of filler tokens emerges. Through this experiment, we observed that during the PT stage, performance with filler tokens was initially lower than with the base input, and only near the end of PT did the two performances become comparable. Then, during the IT stage, performance with filler tokens surpassed the base input, indicating that filler tokens can contribute to additional performance improvement only once the model’s language capabilities reach a sufficiently high level.

---

> ### Author Response · Authors · 2025-11-20
>
> > The attention map analysis hardly explains why the improvements occurred. Strong attention alone does not necessarily translate into better performance. Even so, the paper should present counterexamples (e.g., cases where the model answers incorrectly) and, where possible, quantify these patterns.
>
> Thanks for the good suggestion! We will look for the samples where performance drops and analyze their internal behavior as well.

---

> ### Author Response · Authors · 2025-11-20
>
> > Line 119: The citation(s) should be in parentheses “( … )”.
> Equation (2): Should Z_{1:M} be Z_{T:M}?
>
> Thanks for pointing out the typo, and we’ll fix that!
>
> Equation 2 can be confusing, but as shown in line 158, from the perspective of z, it refers to positions 1 through M (1th and Mth place of z), and from the perspective of x, it is correct to refer to positions T through M.

---

> ### Comment · Reviewer_6CYq · 2025-11-21
>
> "We can’t see what kind of tokens you wrote. They appear as [object Object] vs. [object Object]. Could you please explain in words what kind of tokens they are?"
>
> Sorry for the confusion. I meant [pad] vs [eos].
> It seems that using angle brackets (< >) caused the issue.

---

> ### Comment · Reviewer_6CYq · 2025-11-21
>
> "In our paper, we aimed to investigate how model performance changes when filler tokens are added, and there was no training on the filler tokens themselves. Therefore, the experiments in Section 4.3 were not about training the model with filler tokens, but about testing the hundreds of mid-checkpoints provided by the OLMo team, spanning from the PT to IT stages, to see at which point the effect of filler tokens begins to appear. We were curious about when the performance-boosting effect of filler tokens emerges. Through this experiment, we observed that during the PT stage, performance with filler tokens was initially lower than with the base input, and only near the end of PT did the two performances become comparable. Then, during the IT stage, performance with filler tokens surpassed the base input, indicating that filler tokens can contribute to additional performance improvement only once the model’s language capabilities reach a sufficiently high level."
>
> Yes, I know you didn’t train the models with filler tokens. I just mean that I don’t see any clear trends.

---

### Official Review · Reviewer_V8mp · 2025-10-30

**Soundness:** 2
**Presentation:** 2
**Contribution:** 1
**Rating:** 2
**Confidence:** 4

**Summary:**

This paper investigates whether LLMs can leverage manually inserted filler tokens (such as <pad>, \t, \n ...) at inference time to enhance their reasoning performance. The authors conduct experiments on 6 types of filler tokens on varying sizes of LLMs, revealing even filler tokens can lead to performance variations, especially for small size LLMs.

**Strengths:**

1. The writing and structure of this paper are clear and easy to understand.
2. The research question is interesting. It would be attractive if we find that this approach can consistently boost the performance of LLMs through simple prompt manipulation.

**Weaknesses:**

1. The main weakness of this paper is that the performance of LLMs after injecting filler tokens varies. Some LLMs perform better on MMLU and ARC after injecting filler tokens, while others do not. The authors neither explain nor analyze this phenomenon (Section 4.4 reads more like a case study than a rigorous analysis), nor do they make efforts to avoid performance degradation in a training-free manner. As a result, the proposed approach appears to add random manipulation to the input prompt rather than serving as a convincing, beneficial algorithm that can consistently enhance LLM performance.
2. Although the authors show that the performance of OLMo-2 can be enhanced by filler-token fine-tuning in Section 4.3, it requires large-scale pretraining and instruction tuning, while the performance gain is marginal (~1% in Figure 5) compared to the heavy computational cost. This also somewhat deviates from the main goal of the paper, which is enhancing LLM performance by expanding computation space at inference time. Furthermore, this validation is only conducted on one LLM with one filler token, making it unclear whether the training can generalize.
3. Typo: in Line 218, GSM8K contains 1319 test set examples, not 1819.

**Questions:**

1. What is the training corpus in Section 4.3?

---

> ### Author Response · Authors · 2025-11-20
>
> We appreciate the reviewer for reading our paper and giving valuable feedback, and you can find our answers below:
>
> > The main weakness of this paper is that the performance of LLMs after injecting filler tokens varies. Some LLMs perform better on MMLU and ARC after injecting filler tokens, while others do not. The authors neither explain nor analyze this phenomenon (Section 4.4 reads more like a case study than a rigorous analysis), nor do they make efforts to avoid performance degradation in a training-free manner. As a result, the proposed approach appears to add random manipulation to the input prompt rather than serving as a convincing, beneficial algorithm that can consistently enhance LLM performance.
>
> According to our experimental results, larger models achieve higher baseline performance (without filler tokens) because they can fully leverage their sufficiently large representational space. The reason SmolLM shows the most pronounced change in performance is that its representational space is small; due to the lack of space needed to solve the task, providing filler tokens leads to a sharp performance boost. However, our input manipulation is an unusual form of input change that was not considered during the PT/IT stages of each model, so the ability to exploit it varies greatly depending on the model type and size. As shown in previous work, where training with filler tokens can yield performance comparable to CoT, we expect that additional training on the input format we introduce would lead to even greater improvements. Nevertheless, the scope of our study is to show that models already possess the ability to exploit such input patterns even during inference, despite never having seen them during training.
>
> We were also cautious about the possibility that even small prompt variations could change a model's output. Therefore, to verify that the effects of filler tokens were consistent, we conducted experiments with three different seeds. The results were very similar across seeds, and we reported the averaged performance. This shows that the effect of filler tokens is not random but exhibits a similar pattern across different seeds.

---

> > ### Comment · Reviewer_V8mp · 2025-11-28
> >
> > Thanks to the authors for the detailed responses. However, my concerns are still not well addressed.
> >
> > In other words, my main concern in the original review is about **the value and applicability of this study**. I understand the authors’ claims that *“the scope of our study is to show that models already possess the ability to exploit such input patterns even during inference”* and that *“our paper aims to understand how a model’s performance changes when filler tokens are inserted into the input, and at no point was the model trained on filler tokens.”* However, the current experimental results of this study do not yet seem applicable or insightful to the community. The reasons are twofold:
> > 1. Directly inserting filler tokens at inference time leads to inconsistent performance fluctuations, which cannot provide clear guidance for subsequent researchers or users seeking performance gains. Given the fluctuating performance after inserting filler tokens, I also wonder whether similar fluctuations would occur if we instead inserted some noise text (e.g., “aaaaa” or meaningful but irrelevant text) at inference time.
> > 2. If the authors could demonstrate that current LLMs can be significantly enhanced by lightweight fine-tuning on filler tokens, I would consider this study more applicable and beneficial. However, the experiments in Figure 4 and Figure 5 show that the required training is highly computationally demanding while the performance gains are marginal, which makes me believe that the value of this study still needs further improvement.
> >
> > Therefore, I prefer to keep my original score.

---

> ### Author Response · Authors · 2025-11-20
>
> > Although the authors show that the performance of OLMo-2 can be enhanced by filler-token fine-tuning in Section 4.3, it requires large-scale pretraining and instruction tuning, while the performance gain is marginal (~1% in Figure 5) compared to the heavy computational cost. This also somewhat deviates from the main goal of the paper, which is enhancing LLM performance by expanding computation space at inference time. Furthermore, this validation is only conducted on one LLM with one filler token, making it unclear whether the training can generalize.
>
> First, our paper aims to understand how a model’s performance changes when filler tokens are inserted into the input, and at no point was the model trained on filler tokens. Therefore, the experiment described in Section 4.3 was not about training the model with filler tokens, but about examining hundreds of mid-checkpoints (released by the OLMo team) spanning the entire PT to IT stages, to determine at which point the effect begins to appear when filler-token-containing inputs are provided.
>
> We wanted to identify the stage at which filler tokens start contributing to performance improvements. Through this experiment, we observed that during the PT stage, performance with filler tokens is lower than with the standard input, and it is only near the end of PT that the two performances become similar. Then, in the IT stage, we found that performance with filler tokens surpasses that of the standard input. This indicates that filler tokens can lead to additional performance gains only after the model’s linguistic capabilities have reached a sufficiently advanced level.

---

> ### Author Response · Authors · 2025-11-20
>
> > Typo: in Line 218, GSM8K contains 1319 test set examples, not 1819.
>
> Thank you for catching the typo. The correct number is 1,319, and we will make the revision.

---

> ### Author Response · Authors · 2025-11-20
>
> > What is the training corpus in Section 4.3?
>
> In the experiment in Section 4.3, we measured performance at each mid-checkpoint of the PT model, using checkpoints provided by OLMo rather than models we trained ourselves (https://huggingface.co/allenai/OLMo-2-1124-13B/tree/main
> ). As shown in Figure 4, we evaluated performance from the earliest model parameter checkpoint in Stage 1 through the checkpoints in Stage 2. Stage 1 used the OLMo 2 Mix 1124 model trained on datasets collected from the web, including text, code, and math, while Stage 2 used a model trained on the “ingredient 4” dataset, which reportedly achieved the best performance in the math domain.

---

### Official Review · Reviewer_N9o6 · 2025-11-03

**Soundness:** 2
**Presentation:** 3
**Contribution:** 2
**Rating:** 2
**Confidence:** 4

**Summary:**

The work explore the computational spaces in inference time to see the impact of the inserted tokens (space, enter, tab, period, etc) and the number of such inserted tokens. The author additionally attempted to show the attention distributions to show how the LLM innerworkings get impacted by such insertions.

**Strengths:**

Rich experiments
- models from SmolLM to Gemma to Llama to Qwen to OLMo.
- good dataset selection - ARC and MMLU - that have different domain QAs.

Meaningful results
- token types (e.g., pad vs space), numbers while inferencing
- attention map (fig. 6) shows meaningful results with the high scores with filling tokens/ question-context pairs (though it might have been cherry-picked results)
- interesting that the enter token plays a meaningful role in inferring better responses in Instruct prompts settings (fig. 5) regardless of the number of the tokens.

**Weaknesses:**

The number of tokens
- Table 1 - seems not informative; isn't this obvious that the performance decreases if the number of tokens increase extremely a lot (e.g., 2048 or more), blocking models to use the meaningful tokens (given that current LLMs struggle with long context reasoning, this is what we can expect). Performances with the numbers below this seem to be similar (very negligible changes) c with those of the baselines (without additional tokens at all).
- Only meaningful changes are on SmolLM (1.7B) but not sure this can be a general pattern across models.
- the OLMo performances with increased number of steps (figs. 4 and 5) don't seem to show meaningful results

The filler tokens (+ numbers)
- Fig 2 - this shows some complementary results for Table 1 in regards to the performance changes from number of tokens. One weak point is that they don't seem to have meaningful patterns that share across datasets or models? despite interesting performance changes with certain tokens such as space. and those results were pretty known findings, not limited to this work [A]. so it would have been more interesting if there were more meaningful contributions.



[A] The Butterfly Effect of Altering Prompts: How Small Changes and Jailbreaks Affect Large Language Model Performance, ACL findings 2024

**Questions:**

my main concern for this work is that the contribution seems weak - the number of tokens, and the type of tokens do not seem to show general patterns that adds meaningful findings to the community. Would the authors can clarify them better in case I missed something or undermined the points?

I've also expected the work clarifies how those inserted tokens expand the computational search space for the models to infer responses better - but I couldn't anything regarding this. Could you point me toward those relevant information?

---

> ### Author Response · Authors · 2025-11-20
>
> Thank you for your careful reading and suggestions!
>
> Please find our answers below:
>
> > Table 1 - seems not informative; isn't this obvious that the performance decreases if the number of tokens increase extremely a lot (e.g., 2048 or more), blocking models to use the meaningful tokens (given that current LLMs struggle with long context reasoning, this is what we can expect).
>
> We also agree that this is, to some extent, an expected outcome, as you mentioned. When the model’s input becomes longer, the *lost-in-the-middle* issue arises, where the model fails to properly reference meaningful tokens. However, we conducted this experiment to report at which approximate input length the performance degradation becomes abrupt (like the reviewer 3eZp, the readers might be curious about the results), and since it was not our main result, we included it in the appendix.

---

> ### Author Response · Authors · 2025-11-20
>
> > Performances with the numbers below this seem to be similar (very negligible changes) c with those of the baselines (without additional tokens at all).
> Only meaningful changes are on SmolLM (1.7B) but not sure this can be a general pattern across models.
>
> Based on our experimental results, larger models inherently achieve higher baseline performance without filler tokens, because they can fully utilize their sufficiently large representational space (and it is obvious that larger models generally perform better). The reason SmolLM shows the most pronounced change in performance might be that its representational space is relatively small; due to the limited space available for solving the task, the model exhibits a sharp performance gain when filler tokens are provided.
>
> However, our input manipulation is an unusual type of perturbation that was not considered during the PT/IT stages of any model, so the ability to exploit it varies significantly across both model types and sizes. As shown in previous work [1], when models are trained with filler tokens, they can achieve performance comparable to CoT. Similarly, we expect that training on the input pattern we introduced would lead to even larger improvements.
>
> Nevertheless, the scope of our study is to demonstrate that models can exploit this expanded input space during inference, despite not being trained in this manner.
>
>
> [1] Let's Think Dot by Dot: Hidden Computation in Transformer Language Models

---

> ### Author Response · Authors · 2025-11-20
>
> > the OLMo performances with increased number of steps (figs. 4 and 5) don't seem to show meaningful results
>
> We wanted to check at which stage of model training the ability to utilize filler tokens begins to emerge, so we evaluated checkpoints from the very beginning of PT through the completion of IT, and reported the results in Figures 4 and 5. Our input (solid line) shows a performance trend similar to the no–filler-token condition (dashed line). This indicates that our methodology depends on the model’s inherent capabilities. Since our goal was not to train the model on filler-token inputs but to examine whether the model itself can make use of them, this outcome is expected.
>
> What is noteworthy is that, during the PT phase, the performance generally aligns with the model’s baseline but fluctuates, whereas in the IT phase, although the trend is similar, it consistently outperforms the baseline. From this, we conclude that once the model’s linguistic abilities reach a certain level of stability, it begins to acquire the ability to leverage such filler tokens.

---

> ### Author Response · Authors · 2025-11-20
>
> > Fig 2 - this shows some complementary results for Table 1 in regards to the performance changes from number of tokens. One weak point is that they don't seem to have meaningful patterns that share across datasets or models? despite interesting performance changes with certain tokens such as space. and those results were pretty known findings, not limited to this work [A]. so it would have been more interesting if there were more meaningful contributions.
> [A] The Butterfly Effect of Altering Prompts: How Small Changes and Jailbreaks Affect Large Language Model Performance, ACL findings 2024
>
> As shown in prior work [A] you mentioned, even small changes in the prompt can alter a model’s response. Therefore, to verify whether the effects we observed with filler tokens were consistent, we conducted experiments using three different seeds. The results were very similar across seeds, and we reported the averaged performance. This indicates that the effect of filler tokens is not random but exhibits a similar pattern across different seeds.

---

> ### Author Response · Authors · 2025-11-20
>
> > 1) my main concern for this work is that the contribution seems weak - the number of tokens, and the type of tokens do not seem to show general patterns that adds meaningful findings to the community. Would the authors can clarify them better in case I missed something or undermined the points?
>
> Our study does not propose a method for improving reasoning. Rather, it investigates whether LLMs possess the ability to make use of seemingly meaningless filler tokens—even when such inputs were never encountered during PT or IT.
>
> Although this input format was never seen during training, we investigated how an expanded input space affects model performance. This was something that prior work could not reveal, and we examined whether models improve when given such expanded input space, as well as which token types influence performance for different model sizes. Our findings show that, despite appearing useless, the expanded space actually directs attention to important parts of the input—as seen in the attention maps—and this phenomenon indeed contributes to performance improvements.
>
> By evaluating checkpoints from the very beginning to the end of PT and IT, we confirmed that the performance of this input format is fundamentally influenced by the model’s inherent capability, but becomes beneficial once the model has acquired a certain level of linguistic proficiency (after PT). These results suggest that, similar to prior work, if models were trained on this type of input format, they could achieve even stronger performance.

---

> ### Author Response · Authors · 2025-11-20
>
> > 2) I've also expected the work clarifies how those inserted tokens expand the computational search space for the models to infer responses better - but I couldn't anything regarding this. Could you point me toward those relevant information?
>
> How inserted tokens expand the model’s computational space can be observed through the attention map examples. In Figure 6, the lower triangular region corresponds to the inserted tokens, and we can see cases where high attention scores are directed toward important parts of the actual input. Moreover, different attention heads assign attention to different aspects: tokens representing answer candidates such as A, B, C, and D; the parts of the input that provide crucial hints; and functional elements of the input such as eot. This shows that inserted tokens are not meaningless—rather, they help the model make important decisions within the input.
>
> Similarly, in Figure 9, even when as many as 64 tokens are inserted (the lower triangular region), we can still observe that the model attends to the key parts of the input (bright sky-blue, white, and pale-colored regions).

---

### Author Response · Authors · 2025-12-03

Thank you very much for taking the time to read our manuscript and for providing your valuable feedback. The phenomenon we observed is highly unusual; however, as many of you pointed out, the inconsistency of the results across different model families and sizes appears to be a weakness of our study. We will conduct additional experiments to investigate further which factors in the expanded space may have led to such an unusual outcome. We sincerely appreciate the reviewers’ insightful comments once again.

---

### Meta-Review · Area_Chair_ubSg · 2025-12-09

**Summary:**

This paper presents an empirical study. Despite conducting numerous experiments, the results across different models and scales are inconsistent. Consequently, the empirical analysis fails to yield meaningful or insightful conclusions. Furthermore, in the absence of clear findings, the motivation for the work remains unclear. Finally, given the purely empirical nature of the study, its technical contributions are limited.

**Reviewer Concerns:**

The authors have not addressed the major concerns raised by the reviewers. A summary of these outstanding issues is provided in the section above.

**Reviewer Scores:**

The authors' rebuttals are unconvincing. Most of the reviewers' concerns remain unaddressed, and I do not believe the four reviewers would revise their scores based on the current response.

---

### Decision · Program_Chairs · 2026-01-26

Reject